# Optically oriented attachment of nanoscale metal-semiconductor heterostructures in organic solvents via photonic nanosoldering

Matthew J. Crane [1,7], Elena P. Pandres[1,7], E. James Davis[1], Vincent C. Holmberg [1,2,3]* & Peter J. Pauzauskie[1,2,4,5,6]*

As devices approach the single-nanoparticle scale, the rational assembly of nanomaterial heterojunctions remains a persistent challenge. While optical traps can manipulate objects in three dimensions, to date, nanoscale materials have been trapped primarily in aqueous solvents or vacuum. Here, we demonstrate the use of optical traps to manipulate, align, and assemble metal-seeded nanowire building blocks in a range of organic solvents. Anisotropic radiation pressure generates an optical torque that orients each nanowire, and subsequent trapping of aligned nanowires enables deterministic fabrication of arbitrarily long heterostructures of periodically repeating bismuth-nanocrystal/germanium-nanowire junctions. Heat transport calculations, back-focal-plane interferometry, and optical images reveal that the bismuth nanocrystal melts during trapping, facilitating tip-to-tail "nanosoldering" of the germanium nanowires. These bismuth-semiconductor interfaces may be useful for quantum computing or thermoelectric applications. In addition, the ability to trap nanostructures in oxygen- and water-free organic media broadly expands the library of materials available for optical manipulation and single-particle spectroscopy.

[1] Department of Chemical Engineering, University of Washington, Seattle, WA 98195-1750, USA. [2] Molecular Engineering & Sciences Institute, University of Washington, Seattle, WA 98195-1652, USA. [3] Clean Energy Institute, University of Washington, Seattle, WA 98195-1653, USA. [4] Department of Materials Science and Engineering, University of Washington, Seattle, WA 98195-2120, USA. [5] Institute for Nano-Engineered Systems, University of Washington, Seattle, WA 98195-1653, USA. [6] Physical and Computational Sciences Directorate, Pacific Northwest National Laboratory, Richland, WA 99352, USA. [7] These authors contributed equally: Matthew J. Crane, Elena P. Pandres. *email: holmvc@uw.edu; peterpz@uw.edu

The ability to synthesize nanoscale colloidal materials with controlled composition, morphology, and electronic structure has increased rapidly over the past decades[1]. However, our capacity to integrate nanomaterials into devices remains limited by the ability to deterministically assemble atomically precise nanostructures[2–7]. Creating junctions between different nanomaterials is a unifying challenge in the production of almost all nanoscale devices, and overcoming barriers in this area will disrupt conventional technologies and spur the development of new ones[8–13]. Devices built from colloidal nanomaterials that rely on rectifying heterostructures—e.g. highly efficient photovoltaics[14,15], light emitting diodes[16,17], and sensors[18–20]—are typically produced by sequential deposition of different colloidal nanoparticle solutions to form junctions between layers of materials via spin coating, dip coating, or doctor blading[21]. In addition, there are a range of techniques to assemble colloidal materials in solution based on hydrodynamic effects or particle–particle interactions[22–25]. However, none of these methods deterministically produce heterojunctions between individual nanoparticles. An alternative approach is to directly synthesize periodic nanowire junctions, for example, by exchanging gaseous semiconductor precursors during a seed particle-directed growth process to yield epitaxially matched nanowire heterojunctions[26,27]. Nonetheless, this approach is limited by the compatibility of the seed particle with the semiconductor material. Moreover, the above methods cannot produce rationally designed metal–semiconductor–metal or metal–metal–metal junctions between colloidal nanomaterials, critical for quantum computing and thermoelectrics[13]. For example, superconductor/ non-superconductor interfaces, such as those in a superconductor–bismuth–superconductor junction, are important for quantum computing via Majorana states and studies of topological insulators[11,12]. Currently, applications involving single- or few-particle junctions are forced to rely on lithography due to the challenges outlined above. However, lithographically synthesized nanomaterials are limited in comparison to colloidal chemistries, which can be used to produce materials with a wide range of compositions, geometries, and dopant distributions.

In 2018, Arthur Ashkin shared the Nobel Prize in Physics for the development of the optical trap, which allows the three-dimensional, non-contact manipulation and study of individual nanostructures[2]. Optical traps have the potential to deterministically manufacture heterojunctions between individual nanostructures[1,3,4,9,10,28–31]. To date, the use of optical traps has almost exclusively been limited to the study of cells or materials dispersed in aqueous media[32,33]. The restriction to aqueous solvents severely limits both the library of materials available for single particle trapping and the range of possible experiments. For example, a large fraction of technologically important colloidal nanomaterials are produced and handled in anhydrous organic solvents, as exposure to water can often interfere with the desired chemistry or degrade material properties. Moreover, exclusively trapping in water limits studies of hydrodynamic effects, chemical reactions, and electronic excitations to the range of physical properties exhibited by water, water-soluble reactants, and electronic transitions that will not be quenched by dipole coupling to water[1].

Although nonaqueous solvents have been explored as media for optical trapping in a few reports—for example, Spadaro et al. demonstrated optical trapping of gold–silica mesocapsules in ethanol[34], Sainis et al. investigated the impact of surface moieties on organic beads in nonpolar solvents[35], and Black et al. presented the optical trapping of silica-coated polymer microspheres in organic solvents[8]—to the best of our knowledge, the optical trapping and manipulation of colloidal inorganic nanomaterials in organic solvents has not yet been reported. Here, we demonstrate the optical trapping and assembly of metal nanocrystal-seeded semiconductor nanowires in organic solvents with viscosities up to 28 times greater than water. A modified version of an established organic solution-liquid-solid growth process was employed to synthesize bismuth-seeded germanium nanowires[36–38], which were then used as nanoscale building blocks for the optical assembly of extended semiconductor–metal–semiconductor heterostructures in organic solution. Both discrete dipole approximation (DDA) calculations and experimental data show that the bismuth nanocrystal seeds experience greater radiation pressure than the germanium nanowires, causing the bismuth-tipped ends of the nanowire to orient away from trapping laser, with the nanowire's growth axis aligned parallel to the Poynting vector of the incident laser[39,40]. We leveraged this effect in tandem with an optically driven nano-soldering process, to construct arbitrarily long, periodic bismuth nanocrystal-germanium nanowire heterostructures freely in non-aqueous solution. Heat transport analysis reveals significant photothermal heating of optically trapped nanowires in organic solvents relative to aqueous environments due to the reduced thermal conductivity of the organic solvent. Moreover, coupled electromagnetic and heat transport simulations demonstrate that particles trapped and assembled freely in solution reach significantly higher local temperatures—without bubble nucleation or autoignition—as compared to particle trapping and assembling on a surface, due to the large Young–Laplace interfacial surface pressure[32,41].

## Results

**Optical trapping and alignment in organic solvents.** To trap, manipulate, and assemble colloidal nanostructures in organic solvents, we employed a home-built optical trap (Fig. 1a) that consisted of a tunable, linearly polarized, diode-pumped, solid-state $Yb^{3+}$: YAG thin-disk laser set to 1020 nm, a beam expansion region, a 100X trapping oil-immersion objective (NA = 1.25), and a modified perfusion chamber with a 300-µm spacer[33,42]. Germanium nanowires were grown from colloidal bismuth nanocrystal seeds[36–38] via a solution-liquid-solid process using a modified version of an established synthetic protocol (Supplementary Note 1)[43]. The perfusion chamber was loaded and sealed in a nitrogen-filled glovebox. The nanowires investigated in this study had monodisperse diameters tunable from 10 to 100 nm with corresponding lengths ranging from 0.1 to 5 µm, depending on the chosen synthetic parameters. The bright-field and dark-field transmission electron microscopy (TEM) images in Fig. 1b, c illustrate the presence of bismuth nanocrystals on the tips of the germanium nanowires after growth, as well as the monodispersity of the nanowires used in the optical trapping experiments (Supplementary Fig. 8).

To demonstrate the range of solvents available for optical trapping in organic media, we prepared nanowire dispersions in mixtures of toluene and squalane, as shown in Supplementary Movie 1. This significantly expands the currently reported range of liquid viscosities suitable for trapping from 1 cP to 28.4 cP with Prandtl numbers ranging from 7.1 to 317[1,8,44]. We were able to trap individual nanowires at powers ranging from 1.0 to 10 W (limited by the trapping laser). Figure 2a shows the power spectrum of an optically trapped germanium nanowire in squalane, illustrating the Hookean trapping force acting on the nanowire. As discussed below, power-dependent temperatures and highly non-isothermal temperature distributions within optically trapped particles prevent an exact determination of the trap stiffness. With this caveat in mind, assuming a non-isothermal particle and extrapolating solvent properties (Supplementary Fig. 6), a trapping stiffness of ~1.0 pN µm$^{-1}$ is calculated. Notably, these trapping stiffnesses are comparable to those observed for InP nanowires trapped in water[45,46]. We observed that nanowires longer than 1 µm were generally easier to trap than shorter nanowires. To achieve stable trapping, the

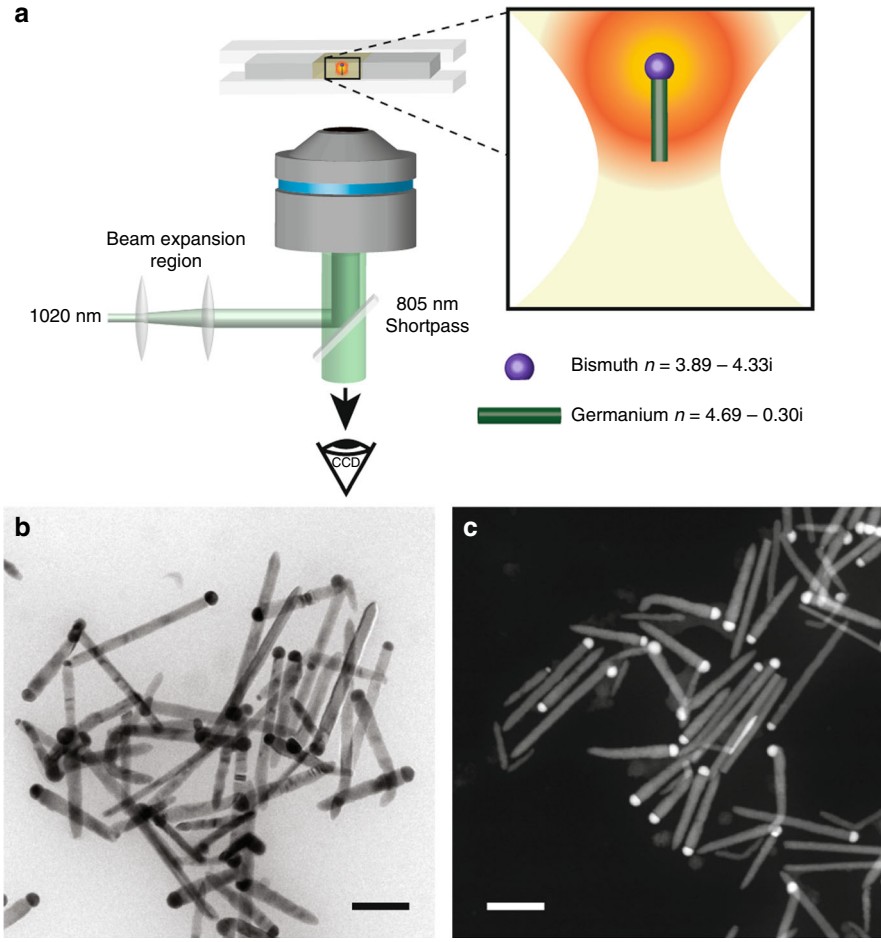

**Fig. 1** A diagram of the optical trap and images of bismuth-seeded germanium nanowires. A schematic of the optical trapping experiment (**a**), and bright-field TEM (**b**) and high-angle annular dark field scanning TEM (HAADF-STEM) (**c**) images of the germanium nanowires used in this study. Scale bars are 200 nm

gradient force must eventually overcome the scattering force. The potential well created by the combination of these forces establishes the equilibrium position of the optically trapped particle, which typically sits slightly above the focal plane of the trap. One potential explanation for the difficulty of trapping smaller wires could be a polarization-induced torque in the optical trap that draws the nanowire into the plane of linear polarization. Tilting the nanowire increases its projected surface area and the scattering force, thus prohibiting trapping[47]. The observation that longer nanowires trapped more reliably could also be due to the different optical gradient and scattering forces acting on the germanium nanowire and the attached bismuth nanocrystal, which act to repel the bismuth tip. To estimate these forces, we used Discrete Dipole Scattering (DDSCAT) code[39] to calculate the relative radiation pressure on bismuth and germanium within a single nanowire. The radiation pressure exerted on a bismuth nanocrystal was calculated to be ~15 times greater than the radiation pressure on an equivalent volume of germanium[48,49]. The disparate forces experienced by the nanowire and its metal tip are predominantly due to the large imaginary component of the refractive index of bismuth ($n_{Bi} = 3.89-4.33i$), as compared to germanium ($n_{Ge} = 4.65-0.30i$). Thus, in the optical trap, the scattering force is reduced when bismuth sits farther from the focal plane, enabling a more stable optical trap and explaining why longer nanowires trap more efficiently (Supplementary Fig. 7)[46].

These anisotropic radiation-pressure effects also occur prior to optical trapping, as the laser pushes the nanowire toward the focal plane. At large distances from the high-electric-field gradient in the focal plane of the optical trap, radiation pressure is the dominant force, effectively inducing a torque on the nanowire. This torque causes the nanowire to align vertically in the trap with the bismuth nanocrystal-tipped end of the wire oriented away from the laser source (Fig. 2b)[2,47,50,51]. Moreover, thermophoretic[28,52,53] effects due to anisotropic heating of the bismuth nanoparticle and hydrodynamic drag[23,54] are additional forces that act on the nanowire before trapping to push the bismuth end of the nanowire away from the laser source, resulting in nanowire alignment. Given the high free carrier concentration in virtually all metal seeds employed in the growth of semiconductor nanowires, these simulations suggest that anisotropic radiation pressure likely represents a general method to align nanowires produced by seeded-growth mechanisms[7]. As was noted during the optical trapping of similarly high refractive index nanowires in water with lengths that exceed the length of the focal volume (Rayleigh range), the gradient force experienced by the wire was independent of nanowire length[45,51,55]. For germanium nanowires in these experiments, the gradient force acting on the germanium section of nanowire was largely responsible for enabling optical trapping, while the scattering forces experienced by the metal seed were responsible for orientation (Supplementary Fig. 7)[46].

**a**

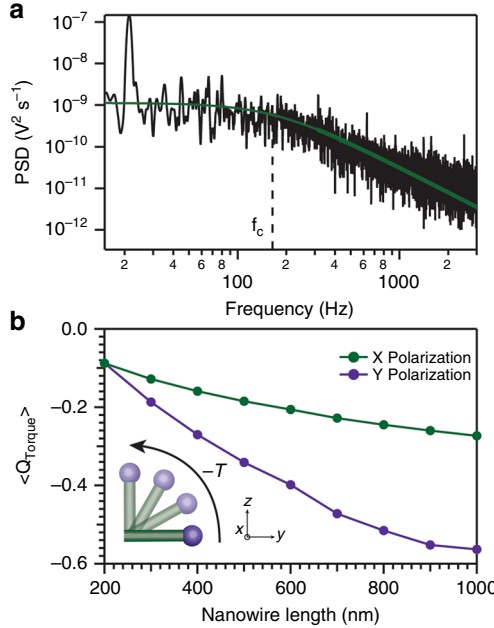

**b**

**Fig. 2** Optical trapping and orientation of metal-seeded nanowires. The power spectrum (**a**) of an optically trapped germanium nanowire was collected via back-focal-plane interferometry and shows radial confinement of the nanowire. The green line shows a fit of the power spectrum to a Lorentzian with a corner frequency of 164 Hz, highlighted by the dashed line. The spike at low frequency is used to calibrate the nanowire position. Prior to optical trapping, far from the focal plane, the nanowire experiences a scattering-induced alignment torque due to the axial anisotropy of the bismuth-tipped nanostructure. Panel **b** illustrates the torque for a y-polarized plane wave averaged over all theta angles of the nanowire

**Nanosoldering Bi nanocrystal–Ge nanowire heterojunctions.** To demonstrate the application of alignment forces in organic solvents, we used the optical trap to assemble periodic bismuth nanocrystal–germanium nanowire heterostructures from colloidal nanowires dispersed in squalane (Fig. 3)[4,9,10,31]. We accomplished this optically oriented alignment and attachment by trapping a single nanowire and then moving it toward a second nanowire in Brownian motion outside the optical trap (Fig. 3a–c). Radiation pressure then oriented and accelerated the second nanowire toward the first trapped nanowire. When the second nanowire reached the first nanowire in the trap, it "soldered" coaxially, tip-to-tail, with the first nanowire, overcoming interfacial electrostatic and hydrodynamic forces (Fig. 3d)[28,30,35,53]. When the nanostructure was released from the trap, optical images showed a bismuth–germanium–bismuth–germanium heterostructure with twice the length of an individual nanowire (Fig. 3e). Re-engaging the optical trap allowed us to repeat this process, soldering additional oriented nanowires to fabricate long, periodic heterostructures in solution (Fig. 3f, Supplementary Movie 2). Radiative heating of the trapped nanostructure (discussed below) enabled the tip-to-tail nanosoldering to produce the repeating nanowire heterostructures. The addition of diphenylgermane (DPG) to the solution enhanced fusion of the optically aligned nanowires. The bismuth seeds are visible at the interface between nanowires in the heterostructure (Fig. 3f). Raman spectroscopy (Fig. 3g) also demonstrates that ensembles of germanium nanowires dispersed in toluene retain their crystallinity even after extended laser heating at high irradiances with a focused (~1.5 MW cm⁻²) and unfocused, collimated (50 kW cm⁻²) beam.

**Laser heating during trapping in organic solvents.** The conventional approach to determine the temperature of an optically trapped particle uses back-focal-plane interferometry to determine the diffusion coefficient of the particle in the trap[56]. With sufficient knowledge of the physical characteristics of the trapping medium and the size of the particle, this information enables calculation of the temperature of the particle[57,58]. Using this formalism, we report both the average temperature at the hydrodynamic radius of trapped nanowires and the average temperature of the nanowire at irradiances relevant for nanosoldering (Fig. 4). These average temperatures vary from 135 to 221 °C at the hydrodynamic radius (Supplementary Note 4) and 266 to 392 °C at the nanowire surface, depending on the trapping power, thus illustrating significant heating of the trapped nanowire, along with a steep temperature gradient in the surrounding fluid. Notably, the temperatures at the nanowire surface are above the bulk bismuth melting point of 271 °C. Moreover, to the best of our knowledge, these temperatures are higher than the temperature of any previously measured, optically trapped nanowire in solution. Finally, although this methodology used to measure particle temperature assumes that the trapped particle is isothermal, the stark differences in the optical properties of the bismuth tip and the germanium wire produce a large thermal gradient within the trapped nanowire that perturbs the local fluid properties.

To validate these temperatures and elucidate the nanosoldering mechanism, we modified the theory developed by Roder et al.[42] for a germanium–bismuth nanowire heterostructure. We approximated the heterostructure as a cylinder with bismuth and germanium sections (Fig. 4a) and solved the heat conduction equation

$$\rho C_p \frac{\partial T}{\partial t} = \kappa \nabla^2 T + \dot{Q}''' \quad (1)$$

in each section of the cylinder, where $\rho$, $\kappa$, and $C_p$ are the density, thermal conductivity, and specific heat capacity, respectively, of the bismuth or germanium portion of the nanowire. The cylinder–cylinder approximation of the heterostructure in Fig. 4a simplifies the heat transport analysis and is valid when the diameter is much smaller than the wavelength of the trapping laser, as in this case. The heat generated per unit time per unit volume, $\dot{Q}'''$, depends on the laser irradiance and polarization, and is related to the electric field **E** and its complex conjugate **E**\* by

$$\dot{Q}''' = \frac{2\pi Re\{n\} Im\{n\}}{\lambda \mu c} \mathbf{E} \cdot \mathbf{E}^*, \quad (2)$$

where $n$ in the complex refractive index, $\lambda$ the wavelength of the incident laser irradiation, $c$ the velocity of light, and $\mu$ the magnetic permeability. Supplementary Table 1 contains all parameter values for the heat transport calculations. For isotropic cylinders and spheres, Mie theory solutions provide the complex internal electric field, **E**, to calculate the source term. However, for the cylinder–cylinder geometry (Fig. 4a), there is no analytical solution. Consequently, we used numerical discrete-dipole calculations to determine the electric field.

As discussed below, photothermal heating melts the bismuth nanocrystal, forming a liquid metal droplet at the nanowire tip that facilitates nanosoldering[43]. As such, we allowed the composition and optical characteristics of the molten droplet to vary with temperature in accordance with the Bi–Ge binary phase diagram. Data regarding nonlinear absorption effects, including saturation and two photon absorption, in bismuth are conflicting, with reported values spanning orders of magnitude. Due to these uncertainties, we created upper and lower bounds by considering heat transport analysis on a germanium nanowire with a bismuth tip and a germanium nanowire without a bismuth tip, respectively. These upper and lower bounds resulted in average temperatures of

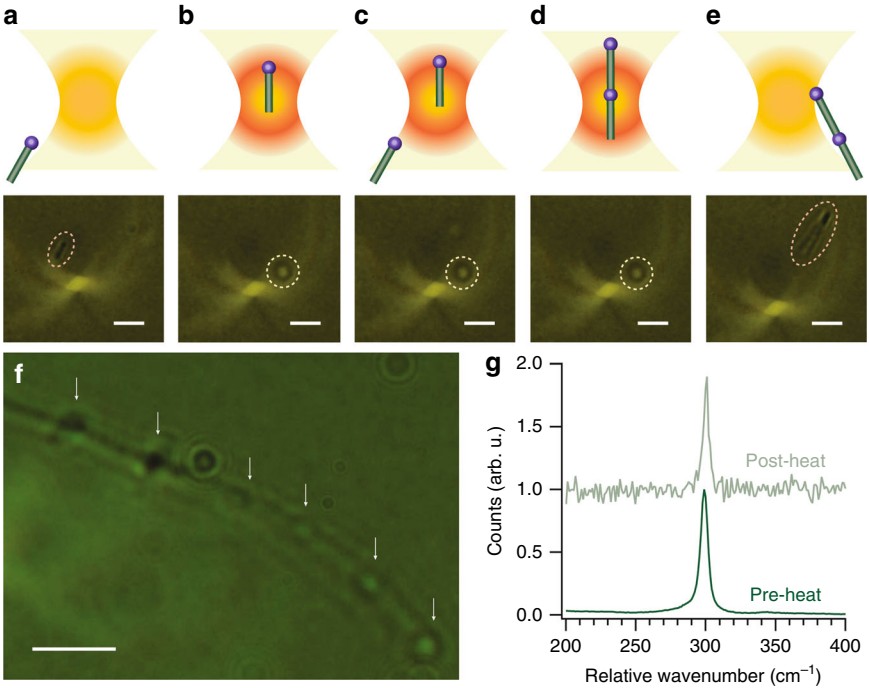

**Fig. 3** Demonstration of optically oriented nanowire assembly via photonic nanosoldering. First, radiation pressure pushes (**a**) a single nanowire into the optical trap (**b**). We then bring the trapped nanowire near a second nanowire diffusing freely in Brownian motion (**c**). Forces due to asymmetric radiation pressure combined with photothermally induced laser heating result in the formation of an optically aligned coaxial heterostructure (**d**), which is subsequently released back into Brownian motion (**e**). An image showing a periodic heterostructure assembled at a trapping power of 5.5 W in an environment of squalane with DPG. The white arrows illustrate bismuth nanocrystal junctions between each of the germanium segments (**f**). Orange circles highlight freely moving, untrapped nanowires while yellow circles indicate optically trapped nanowires. Supplementary Movie 2 illustrates the assembly process. All scale bars are 2 μm. Raman scattering (**g**) from ensembles of germanium nanowires dispersed in toluene before and after photothermal heating by both a focused (~1.5 MW cm$^{-2}$) and unfocused, collimated (50 kW cm$^{-2}$) 1070 nm laser demonstrates that photothermal heating during trapping does not alter the crystallinity of the material

447 °C and 244 °C in an ambient bath of squalane at a trapping power of 5 W, which was used for optically oriented attachment (Fig. 4c).

While the average electric field is similar throughout the nanowire heterostructure, the large value of $Im\{n\} = 4.33$ for bismuth indicates significant absorption relative to $Im\{n\} = 0.30$ for germanium. Furthermore, the low Biot number of the nanowire suggests that there is little radial temperature variation. The radially averaged temperature profile of a single bismuth–germanium nanowire trapped at 5 W (Fig. 4b) illustrates that the bismuth tip is significantly hotter than the germanium wire, due to the disparity in heat generation. We calculate that the bismuth nanocrystal melts at a distance of ~50 μm below the focal plane of the optical trap due to photothermal heating. Indeed, calculations show that a trapping power of only 0.35 W is sufficient to melt the bismuth nanocrystal tip in the optical trap's focal plane. Because the time scale to achieve steady-state heating ($<10^{-9}$ s) is much less than the time for radiation pressure to drive the nanowire into the optical trap ($10^{-2}$ – 1 s), the incoming nanowire is at a steady-state temperature as it approaches the trap[59].

Notably, the trapped nanowire can reach very high temperatures without locally boiling the solvent due to the high Young–Laplace interfacial surface pressure[32]. To demonstrate the impact of the Young–Laplace pressure, we trapped a nanowire in squalane at 5 W and moved it toward a glass slide, which provided a flat surface for bubble nucleation (Supplementary Movie 3)[60]. Although stable in solution, when the nanowire reached the surface of the glass slide, the solvent immediately boiled and ignited, despite the high thermal conductivity of glass compared to squalene, indicating that the large Young–Laplace

pressure prevents local bubble nucleation and ignition near the nanostructure. Importantly, docking with a surface can easily be achieved safely by simply switching to a lower laser power or using a higher viscosity solvent, thus suppressing bubble formation. Both of these strategies enable the safe manipulation of optically trapped materials in organic solvent systems. This serves as a critical safety demonstration, while also highlighting the significant processing advantage afforded by working in solution as opposed to on a surface, thus enabling higher temperature transformations. Moreover, when carrying out a high-temperature manipulation, such as nanosoldering, on a trapped structure in an organic solution, it is critically important to reduce the power of the trapping laser prior to docking the structure with a surface.

## Discussion

This analysis indicates that nanosoldering occurs through sequential steps. First, the radiation pressure from the optical trap acts asymmetrically on the nanowire to exert an optical torque that orients the bismuth nanocrystal tip away from the incident laser and accelerates the nanowire toward a previously trapped and aligned nanowire. When the accelerating nanowire comes within ~50 μm of the trap volume, the temperature of the photothermally heated bismuth nanocrystal exceeds the bismuth melting temperature of 271 °C, rapidly causing the tip of the nanowire to melt and form a liquid metal droplet. As the nanowire continues to approach the optical trap, its temperature rises, and the gradient force begins to oppose the radiation pressure, decreasing the velocity of the nanowire. Finally, as the gradient force slows the incoming, aligned nanowire, the molten bismuth

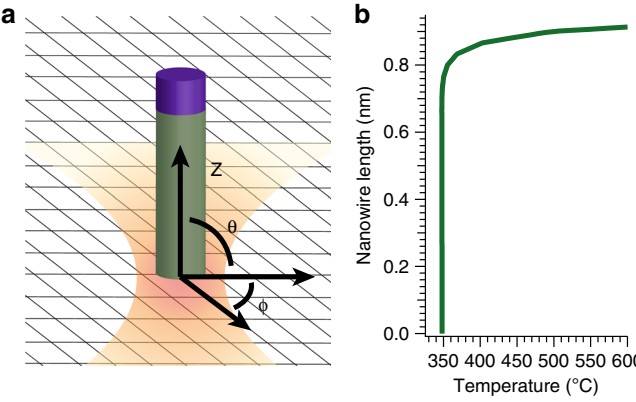

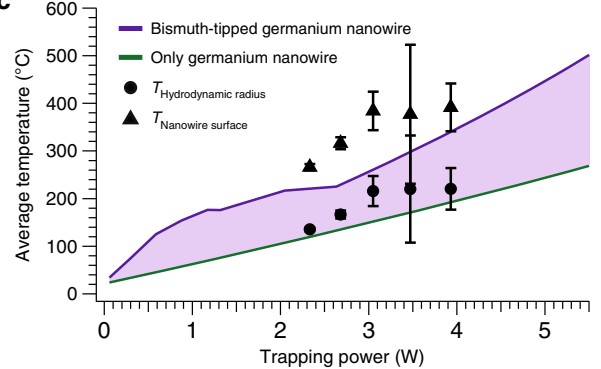

**Fig. 4** Analytically predicted temperatures of an optically trapped bismuth-seeded germanium nanowire. Radially averaged local temperature (**b**) along the length of the nanowire (**a**) during trapping at a power of 5.5 W, showing that the entire nanowire is above the bismuth melting temperature during alignment and nanosoldering, and that the end of the nanowire near the bismuth tip experiences significantly higher local temperatures. To create lower and upper temperature bounds for the optically trapped nanostructure, the average temperatures of trapped germanium nanowires with (purple) and without (green) bismuth tips were calculated as a function of trapping power (**c**). The Supplementary Information includes full details of these calculations. Average experimental temperatures at the nanowire surface were calculated using a hot Brownian motion diffusion model[64]. Error bars indicate the standard deviation of the average temperature from three separate measurements

tip contacts the germanium end of the previously trapped nanowire, fusing the two optically oriented nanostructures together.

In conclusion, we have expanded the range of optical trapping media to include organic solvents and demonstrated the optically oriented assembly of metal-seeded germanium nanowires for the first time. DDA calculations and experimental results show that the bismuth seed particles on the tips of the SLS-grown germanium nanowires experience a significantly greater radiation pressure than the germanium portion of the nanowire, causing it to orient coaxially in the trap, with the bismuth seed oriented away from the incident laser source. In addition, as compared to trapping in aqueous solvents, operating an optical trap in an organic solvent environment facilitates the generation of much higher local temperatures upon trapping due to the decreased thermal conductivity of the solvent. We leveraged these effects to rationally manufacture periodic bismuth nanocrystal–germanium nanowire heterostructures in squalane using an optical trap. We anticipate that these results will usher in a new realm of optical trapping applications, including additive nanomanufacturing from constituent nanomaterial building blocks[9,61], single-particle catalysis[7,8,60], and future hydrodynamic studies[5,6,35,62,63].

## Data availability

The datasets generated during and/or analyzed during the current study are available from the corresponding author on reasonable request.

## Code availability

The codes generated during the current study are available from the corresponding author on reasonable request.

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

## Acknowledgements

This research was supported by the National Science Foundation (NSF) through the UW Molecular Engineering Materials Center, a Materials Research Science and Engineering Center (DMR-1719797), the University of Washington's Molecular Engineering Institute and Institute for Nano-Engineered Systems, and the State of Washington through the University of Washington Clean Energy Institute and via funding from the Washington Research Foundation. Part of this work was conducted at the Molecular Analysis Facility, a National Nanotechnology Coordinated Infrastructure site at the University of Washington, which is supported in part by the National Science Foundation (grant ECC-1542101), the University of Washington, the Molecular Engineering & Sciences Institute, and the Clean Energy Institute. M.J.C. acknowledges a NDSEG Fellowship as well as a WRF Postdoctoral Fellowship. P.J.P. acknowledges support from the Air Force Office of Scientific Research Young Investigator Award (Contract #FA95501210400).

## Author contributions

M.J.C. and E.P.P. performed the experiments and contributed equally to this work. M.J.C., P.J.P. and E.J.D. performed the heat transport calculations. P.J.P. and V.C.H. directed the research. All authors contributed to the design of experiments, scientific discussion, and writing of the manuscript.

## Competing interests

The authors declare no competing interests.
