## [Peer Review File · Nature Communications]

Reviewers' Comments:

Reviewer #1:

Remarks to the Author:

In this paper the Authors describe experiments of optical trapping and manipulation of metal-seeded nanowires in organic solvents. Specifically, they show that composite structures made of a bismuth nanocrystal and a germanium nanowire can align and assemble in the optical trap. The local heating, resulting from the strong absorption of the bismuth particle, promote the soldering and fabrication of a heterostructure with a length of several microns where germanium nanowires are interfaced by bismuth junctions. Accurate heat transport calculations are presented that provide a quantitative understanding of the heating effects on the structures in the trap.

The manuscript reports convincing results on trapping, manipulation, and fabrication of heterostructures in organic solvents. The presented work is novel and I expect to be of interest not only to researchers in the field of optical trapping and manipulation, but to the broader community in material science and nanotechnology. However, there are some important issues that the Authors should address before the manuscript is considered for publication.

Main issues:

1) A major result of the work is the optical trapping of the bismuth-germanium composite particles. However, the Authors do not show any experiments on optical trapping characterization. Optical forces on these composite nanoparticles are only briefly and qualitatively discussed. I would strongly suggest to quantitatively characterize the optical forces acting on the wires. Ideally a QPD tracking of the particles in the trap and a consequent analysis following standard calibration methods can give a lot of information on optical forces in different solvents and the dynamics of these complex particles. Even a simple photodiode can be used to record the trapping of a wire and characterize (estimate) the trap constants. Moreover, the controlled soldering of nanowires into longer structures could be precisely monitored by the change of thermal fluctuations in the trap. The heating transport calculations shown by the Authors can give an accurate estimate of the temperature surrounding the particles and hence on the viscosity of the solvent from which an estimate of the optical forces can be achieved.

2) As pointed out by the Authors, the standard environment for optical trapping experiments is water. Is it possible to contrast the results obtained in organic solvents with experiments in water? Can the composite nanowires be re-dispersed in water or water-diluted samples used to check for similarities/differences in trapping, alignment, fabrication? Or is this not working?

3) Much work has been devoted to the theory and modelling of anisotropic particles in optical traps. Homogeneous nanowires have been used as model particles to understand optical forces and dynamics in the trap [see e.g., Simpson and Hanna. *Nanotechnology* 23 (2012):205502; Borghese et al. *PRL* 100 (2008):163903.]. In the manuscript, lines 115-122, the Authors give only a rough estimate of the radiation pressure difference for a bismuth and a germanium nanoparticle obtained using DDSCAT codes. However, no calculation of the optical trapping forces is considered.

First, I suggest to report the calculated extinction cross sections for the bismuth and germanium particles considered by the Authors and to calculate/compare the corresponding radiation pressure and gradient forces in dipole approximation for the typical parameters (power, beam waist, etc.) used in the experiments.

Then, some simple calculations of optical forces on the bismuth-germanium nanowires can be obtained by using the analytical model developed by Simpson and Hanna [*Nanotechnology*(2012)] for homogeneous nanowires in combination with a simple Maxwell-Garnett mixed dielectric function. Wires of different length will have different dielectric constant because of the different mixing (shorter wires will have a large fraction of bismuth, longer wires a smaller amount instead) and hence different optical gradient and scattering forces can be calculated as a function of length. Despite being an approximated approach, this could give an interesting scaling of the optical forces

on these heterostructures and complement the (expected) information on orientation.

4) Optical trapping of gold-silica mesocapsules in ethanol has been recently demonstrated: Spadaro et al., "Optical Trapping of Plasmonic Mesocapsules: Enhanced Optical Forces and SERS." *The Journal of Physical Chemistry C* 121.1 (2016): 691-700. This reference could be added in the introduction (Lines 74-76) together with Black et al.

Other issues:

5) Lines 97-100: information on the polarization state of the laser is missing. I believe it is linearly polarized. For small nanowires the linear polarization might be a source of instability in the orientation as it competes with the aligning torque caused by the radiation pressure anisotropy due to both the anisotropic composition (bismuth-germanium) and the linear shape (see for example the theoretical analysis shown Borghese et al. *PRL*, 2008). This might be a comment to consider when discussing the instabilities for small wires (Line 114).

6) Lines 105-108: The Authors state that the structures have a monodisperse diameter and a length in the range 0.1-5 microns. Maybe I missed it, but I could not find the value for the measured diameter. I also suggest the Authors to show the distribution of lengths as obtained from the TEM analysis to have a complete characterization of the structures.

7) Line 112: The Authors state that the power used is in the range 1-10 W. It is not clear if this is measured at the sample or at the laser output. If it is measured at the laser output, I suggest to report also the values at the sample.

8) Line 116: "... optical and scattering forces...". I believe the Authors mean "optical gradient and scattering forces" (?)

As mentioned in point 3, in this paragraph there is no discussion about the role of the gradient force. The gradient force is proportional to the real part of the polarizability of the nanoparticles (in a dipole approximation). Hence, a simple analysis of the polarizability could give further insight on the optical trapping behaviour of the composites. For example, the Authors could compare the gradient forces or the trapping constants for nanoparticles with dielectric permittivity of bismuth, germanium or a mixed alloy (using the Maxwell-Garnett).

9) Lines 130-131: "...the gradient force experienced by the wire was independent of nanowire length." This is only true for nanowires longer than the laser spot since the interaction region saturates (see also Simpson, *Nanotechnology* 2012).

10) Line 148: The definition of "DPG" is only given in the Supp. Info. It should be given also in the main text.

Reviewer #2:

Remarks to the Author:

The major claims in this paper cover two phenomena, first is the trapping on nano-wires in non aqueous solvents and the second is the "soldering" of the nanowires together. Put together these two form what is in my opinion an important set of results that is likely to be of significant interest in the field. It is often the case that micro sized components of different materials are needed to create a device; for example various semiconductors to create light emitters or light detectors for sensing or miniature diagnostic devices. However adding these small components with optical tweezers from a water based solution is a problem due to the desire not to immerse the existing circuitry in water. The electronics community is less concerned about other solvents and oils and so the ability to optically tweeze nanowires in other media and attach them to devices will be of great interest.

As the tweezing of nanowires in different solvent is very interesting I would have expected the paper to discuss these results in relation to previously reported tweezing of colloidal particles in nonpolar solvents such as "electrostatic interactions of colloidal particles in nonpolar solvents: role of surface chemistry and charge control agents" in Langmuir 2008, 24, 1160-1164 by Dufresne et al.

The soldering of the nanowires together is also an important finding that will be of great interest to the tweezing community. The findings have good novelty however the physical mechanism that is presented could be more strongly proven as although the authors do give some good literature references and calculations that would suggest their nanowires should heat up there is little physical evidence of the alloy they claim is created. The papers conclusions would be much more convincing if they could show analysis of the soldered nanowires through Raman spectroscopic analysis, EDX, AFM or TEM to show the mix of materials created.

Reviewer #3:

Remarks to the Author:

In this manuscript, Crane et al. demonstrate optical trapping of germanium nanowires in toluene and squalene. It was observed that the nanowires orient with the long axis parallel to the Poynting vector of the trapping laser. Moreover, it was seen that trapping multiple wires results in an end-to-end "nanosoldered" product. The manuscript is well-written and the claims are largely supported by the research conducted.

I believe that TEM images of multiple soldered nanowires are necessary to demonstrate sufficient control over the nanosoldering process yielding a head-to-tail attached product. The authors reason that the germanium nanowires are always trapped with the bismuth seeded end away from the trapping laser due to the differences in complex refractive index between these two materials. This is a non-trivial claim that should be supported by an experimental observable. For example, it is not clear that the difference in scattering between bismuth and germanium is sufficient to flip the orientation of a nanowire that enters the trap with the bismuth seed towards the trapping laser. The Supplementary Movies provided by the authors demonstrate the random nature by which a nanowire approaches the optical trap. Additionally, there is also no practical way to prevent additional objects from "falling into" the optical trap (especially nanoscale objects $\sim < 100\text{nm}$ that are too small to be seen in the real-time images). The practical utility of this method is also potentially hindered by the authors' observation that optical trapped materials tend to explode when brought near the coverslip surface. These concerns can be assuaged if the authors can reproducibly perform the nanosoldering and isolate the products for TEM imaging.

Response to reviewers

Reviewer #1 (Remarks to the Author):

In this paper the Authors describe experiments of optical trapping and manipulation of metal-seeded nanowires in organic solvents. Specifically, they show that composite structures made of a bismuth nanocrystal and a germanium nanowire can align and assemble in the optical trap. The local heating, resulting from the strong absorption of the bismuth particle, promote the soldering and fabrication of an heterostructure with a length of several microns where germanium nanowires are interfaced by bismuth junctions. Accurate heat transport calculations are presented that provide a quantitative understanding of the heating effects on the structures in the trap.

The manuscript reports convincing results on trapping, manipulation, and fabrication of heterostructures in organic solvents. The presented work is novel and I expect to be of interest not only to researchers in the field of optical trapping and manipulation, but to the broader community in material science and nanotechnology. However, there are some important issues that the Authors should address before the manuscript is considered for publication.

Main issues:

1) A major result of the work is the optical trapping of the bismuth-germanium composite particles. However, the Authors do not show any experiments on optical trapping characterization. Optical forces on these composite nanoparticles are only briefly and qualitatively discussed. I would strongly suggest to quantitatively characterize the optical forces acting on the wires. Ideally a QPD tracking of the particles in the trap and a consequent analysis following standard calibration methods can give a lot of information on optical forces in different solvents and the dynamics of these complex particles. Even a simple photodiode can be used to record the trapping of a wire and characterize (estimate) the trap constants. Moreover, the controlled soldering of nanowires into longer structures could be precisely monitored by the change of thermal fluctuations in the trap. The heating transport calculations shown by the Authors can give an accurate estimate of the temperature surrounding the particles and hence on the viscosity of the solvent from which an estimate of the optical forces can be achieved.

We thank the reviewer for their suggestion to include QPD measurements for the manuscript. We conducted QPD measurements of the trapped nanowires and calculated both trap stiffness and temperatures, which are included in Figure 2A, Figure 4C, and Figure S6. The resulting trap stiffnesses and temperatures match literature values, validate the overall heat transport theory, and confirm that the bismuth nanocrystal melts. Including these experiments significantly strengthens the overall manuscript, and we thank the reviewer for these suggestions. We have also included the following discussion regarding the QPD data:

On page 5 and 6: “Figure 2a shows the power spectrum of an optically trapped germanium nanowire in squalane, illustrating the Hookean trapping force acting on the nanowire. As discussed below, power-dependent temperatures and highly non-isothermal temperature

distributions within optically trapped particles prevent an exact determination of the trap stiffness. With this caveat in mind, assuming a non-isothermal particle and extrapolating solvent properties (Figure S6), a trapping stiffness of $\sim 1.0 \text{ pN}\cdot\mu\text{m}^{-1}$ is calculated. Notably, these trapping stiffnesses are comparable to those observed for InP nanowires trapped in water.^{42,43,}

On page 8: “The conventional approach to determine the temperature of an optically trapped particle uses back focal plane interferometry to determine the diffusion coefficient of the particle in the trap. With sufficient knowledge of the physical characteristics of the trapping medium and the size of the particle, this information enables calculation of the temperature of the particle.^{53,54} Using this formalism, we report both the average temperature at the hydrodynamic radius of trapped nanowires and the average temperature of the nanowire at irradiances relevant for nanosoldering (Figure 4). These average temperatures vary from 135 to 221°C at the hydrodynamic radius (defined in the SI) and 266 to 392°C at the nanowire surface, depending on the trapping power, thus illustrating significant heating of the trapped nanowire, along with a steep temperature gradient in the surrounding fluid. Notably, the temperatures at the nanowire surface are above the bulk bismuth melting point of 271°C. Moreover, to the best of our knowledge, these temperatures are higher than the temperature of any previously measured, optically trapped nanowire in solution. Finally, although this methodology used to measure particle temperature assumes that the trapped particle is isothermal, the stark differences in the optical properties of the bismuth tip and the germanium wire produce a large thermal gradient within the trapped nanowire that perturbs the local fluid properties.”

In addition, we have added the relevant methods section to include the QPD experiments and analysis.

2) As pointed out by the Authors, the standard environment for optical trapping experiments is water. Is it possible to contrast the results obtained in organic solvents with experiments in water? Can the composite nanowires be re-dispersed in water or water-diluted samples used to check for similarities/differences in trapping, alignment, fabrication? Or is this not working?

Unfortunately, the germanium nanowires used in this study are inherently hydrophobic in nature, and do not disperse in water. Moreover, even if a ligand exchange was conducted in order to integrate a set of hydrophilic ligands onto the material, germanium is known to rapidly oxidize, corrode, and dissolve in aqueous environments. For example, it has been shown that unpassivated germanium nanowires rapidly oxidize to GeO_x and completely dissolve after only a few hours of exposure to water (Hanrath and Korgel. 2004. *JACS*).

Importantly, these characteristics of germanium are shared by many of the other water-sensitive/oxidation-sensitive nanomaterials that are of interest to the field. These physical properties serve as the fundamental motivation for our study, demonstrating that optical trapping is a viable strategy for the manipulation of many oxygen/water-sensitive materials.

3) Much work has been devoted to the theory and modelling of anisotropic particles in optical traps. Homogeneous nanowires have been used as model particles to understand optical forces and dynamics in the trap [see e.g., Simpson and Hanna. *Nanotechnology* 23

(2012):205502; Borghese et al. PRL 100 (2008):163903.]. In the manuscript, lines 115-122, the Authors give only a rough estimate of the radiation pressure difference for a bismuth and a germanium nanoparticle obtained using DDSCAT codes. However, no calculation of the optical trapping forces is considered.

First, I suggest to report the calculated extinction cross sections for the bismuth and germanium particles considered by the Authors and to calculate/compare the corresponding radiation pressure and gradient forces in dipole approximation for the typical parameters (power, beam waist, etc.) used in the experiments. Then, some simple calculations of optical forces on the bismuth-germanium nanowires can be obtained by using the analytical model developed by Simpson and Hanna [Nanotechnology(2012)] for homogeneous nanowires in combination with a simple Maxwell-Garnett mixed dielectric function. Wires of different length will have different dielectric constant because of the different mixing (shorter wires will have a large fraction of bismuth, longer wires a smaller amount instead) and hence different optical gradient and scattering forces can be calculated as a function of length. Despite being an approximated approach, this could give an interesting scaling of the optical forces on these heterostructures and complement the (expected) information on orientation.

We thank the reviewer for indicating an area of the manuscript where additional calculations would be beneficial. We believe that including these calculations is particularly important given the new solvent environment, which differs significantly from most previous literature, and the optically-induced nanowire alignment. These calculations have led to distinguishing the trapping into two different regimes: far from the trap where the nanowire aligns under only radiation pressure and in the trap where the nanowire experiences both significant radiation pressure and gradient forces.

To address the alignment, we have added a figure showing the angle-dependent, radiation-pressure-induced alignment of the bismuth-tipped germanium nanowire in Figure 2B. These results clearly demonstrate that the radiation pressure acts anisotropically due to the high bismuth extinction cross section relative to germanium extinction cross section, causing the nanowire to align.

We also believe that the composition-dependent trapping gradient force is an interesting idea, and we thank the reviewer for pointing us toward the elegant analytical model developed by Simpson and Hanna, which we have now cited. To address the trapping potential, we have added Figure S7 showing the predicted composition-dependent trapping force.

We have added significant discussion based on these additions as follows:

Page 6 and 7: “The disparate forces experienced by the nanowire and its metal tip are predominantly due to the large imaginary component of the refractive index of bismuth ($n_{Bi} = 3.89 - 4.33i$), as compared to germanium ($n_{Ge} = 4.65 - 0.30i$). Thus, in the optical trap, the scattering force is reduced when the bismuth sits farther from the focal plane, enabling a more stable optical trap and explaining why longer nanowires trap more efficiently (Figure S7).⁴³

These anisotropic radiation-pressure effects also occur prior to optical trapping, as the laser pushes the nanowire toward the focal plane. At large distances from the high electric field gradient in the focal plane of the optical trap, radiation pressure is the dominant force, effectively inducing a torque on the nanowire. This torque causes the nanowire to align vertically in the trap with the bismuth nanocrystal-tipped end of the wire oriented away from the laser source (Figure 2b).^{2,44,47,48} Moreover, thermophoretic^{28,49,50} effects due to anisotropic heating of the bismuth nanoparticle and hydrodynamic drag^{23,51} are additional forces that act on the nanowire before trapping to push the bismuth end of the nanowire away from the laser source, resulting in nanowire alignment.”

In the caption of Figure S7, “Analytical approximation of nanowire longitudinal trap stiffness as a function of length, using the theory developed by Simpson and Hanna.⁴³ To include the effect of nanowire length, we used the Maxwell-Garnett approximation weighed by the volume fraction of the bismuth nanocrystal, which remains constant, and the germanium nanowire, which varies with length. In the first region, the longitudinal trap stiffness increased as the length of that nanowire increased up to ~1200 nm—that is, as the distance from focal plane to the bismuth nanocrystal increased. This reflects the observation that longer nanowires are easier to trap than shorter nanowires. In the second region, the trap stiffness decreased due to a decreased magnitude of the gradient force as more of the nanowire sits outside of the focal plane.”

Combined, we believe that including both of these calculations and relevant discussions address important questions for this new process.

4) Optical trapping of gold-silica mesocapsules in ethanol has been recently demonstrated: Spadaro et al., "Optical Trapping of Plasmonic Mesocapsules: Enhanced Optical Forces and SERS." *The Journal of Physical Chemistry C* 121.1 (2016): 691-700. This reference could be added in the introduction (Lines 74-76) together with Black et al.

We thank the reviewer for directing us to this publication. The reference has been incorporated into the manuscript, along with the following text:

“Although nonaqueous solvents have been explored as media for optical trapping in a few reports—for example, Spadaro *et al.* demonstrated optical trapping of gold-silica mesocapsules in ethanol,³⁴ Sainis *et al.* investigated the impact of surface moieties on organic beads in nonpolar solvents,³⁵ and Black *et al.* presented the optical trapping of silica-coated polymer microspheres in organic solvents⁸—to the best of our knowledge, the optical trapping and manipulation of colloidal inorganic nanomaterials in organic solvents has not yet been reported.”

Other issues:

5) Lines 97-100: information on the polarization state of the laser is missing. I believe it is linearly polarized. For small nanowires the linear polarization might be a source of instability in the orientation as it competes with the aligning torque caused by the radiation pressure anisotropy due to both the anisotropic composition (bismuth-germanium) and the linear shape (see for example the theoretical analysis shown Borghese et al. PRL, 2008). This might be a comment to consider when discussing the instabilities for small wires (Line 114).

The laser is indeed linearly polarized. This information has been added to the manuscript, and we thank the reviewer for pointing out this oversight.

The reviewer also makes an excellent point with respect to the linear polarization being a potential source of instability for small wires. We have added the following discussion to the manuscript on page 6, and incorporated the Borghese *et al.* reference, as suggested:

“To achieve stable trapping, the gradient force must eventually overcome the scattering force. The potential well created by the combination of these forces establishes the equilibrium position of the optically trapped particle, which typically sits slightly above the focal plane of the trap. One potential explanation for the difficulty of trapping smaller wires could be a polarization-induced torque in the optical trap that draws the nanowire into the plane of linear polarization. Tilting the nanowire increases its projected surface area and the scattering force, thus prohibiting trapping.⁴⁴”

6) Lines 105-108: The Authors state that the structures have a monodisperse diameter and a length in the range 0.1-5 microns. Maybe I missed it, but I could not find the value for the measured diameter. I also suggest the Authors to show the distribution of lengths as obtained from the TEM analysis to have a complete characterization of the structures.

We thank the reviewer for pointing out this missing information. Notably, the nanowire length can be tuned by adjusting the semiconductor precursor to metal nanocrystal seed ratio used in a particular synthesis (Chockla *et al.* 2011. *Chem. Mater.*), while the nanowire diameter can be tuned by choosing both the reaction temperature (Chockla *et al.* 2011. *Chem. Mater.*) and the diameter of the metal nanocrystal seeds (Hanrath and Korgel. 2002. *JACS*). While nanowire diameters typically range from 10 to 100 nm using this particular synthetic strategy, for a given synthesis, diameters are quite monodisperse. See for example Chockla *et al.* (2011), where nanowires with lengths of 170 ± 20 nm exhibited diameters of 11.4 ± 1.9 nm.

As suggested by the reviewer, we have now incorporated additional characterization data into the Supporting Information, with TEM images showing the nanowire length and diameter distributions, as well as the associated length and diameter histograms.

Moreover, we have also modified Figure 1 to include a characteristic bright-field and dark-field TEM images that clearly illustrate the dimensions of the nanowires used in this study, and also highlight the distinct bismuth and germanium sections of each nanostructure.

The statement in the text has been modified as follows:

“The nanowires investigated in this study had monodisperse diameters tunable from 10 to 100 nm with corresponding lengths ranging from 0.1 to 5 micrometers, depending on the chosen synthetic parameters. The bright-field and dark-field transmission electron microscopy (TEM) images in Figure 1b,c illustrate the presence of bismuth nanocrystals on the tips of the germanium nanowires after growth, as well as the monodispersity of the nanowires used in the

optical trapping experiments (See Supporting Information for length and diameter distributions).”

7) Line 112: The Authors state that the power used is in the range 1-10 W. It is not clear if this is measured at the sample or at the laser output. If it is measured at the laser output, I suggest to report also the values at the sample.

We thank the author for noting this omission. We have added a short description in the methods section describing how the laser power was measured, on page 26: “The powers reported in the manuscript are measured after beam expansion and before the high numerical aperture trapping objective, which decreases the laser power by less than 30% at these trapping wavelengths.”

8) Line 116: “... optical and scattering forces...”. I believe the Authors mean “optical gradient and scattering forces” (?)

This is correct. The statement has been revised, as suggested by the reviewer.

As mentioned in point 3, in this paragraph there is no discussion about the role of the gradient force. The gradient force is proportional to the real part of the polarizability of the nanoparticles (in a dipole approximation). Hence, a simple analysis of the polarizability could give further insight on the optical trapping behaviour of the composites. For example, the Authors could compare the gradient forces or the trapping constants for nanoparticles with dielectric permittivity of bismuth, germanium or a mixed alloy (using the Maxwell-Garnett).

We thank the reviewer for noting that we have not explicitly discussed the relationship between the gradient and scattering forces. Similarly, we believe that a short discussion on the relationship between these two will aid general readers, making the manuscript more accessible to a wider audience. In response, we have added Figure S7, illustrating the effect of the nanowire’s average polarizability (calculated via the Maxwell-Garnett approximation) on trapping forces. The text was also updated as follows:

On page 6, “To achieve stable trapping, the gradient force must eventually overcome the scattering force. The potential well created by the combination of these forces establishes the equilibrium position of the optically trapped particle, which typically sits slightly above the focal plane of the trap.”

On page 6, “The disparate forces experienced by the nanowire and its metal tip are predominantly due to the large imaginary component of the refractive index of bismuth ($n_{Bi} = 3.89 - 4.33i$), as compared to germanium ($n_{Ge} = 4.65 - 0.30i$). Thus, in the optical trap, the scattering force is reduced when the bismuth sits farther from the focal plane, enabling a more stable optical trap and explaining why longer nanowires trap more efficiently (Figure S7).⁴³”

In Figure S7’s caption, “Analytical approximation of nanowire longitudinal trap stiffness as a function of length, using the theory developed by Simpson and Hanna.⁴³ To include the effect of

nanowire length, we used the Maxwell-Garnett approximation weighed by the volume fraction of the bismuth nanocrystal, which remains constant, and the germanium nanowire, which varies with length. In the first region, the longitudinal trap stiffness increased as the length of that nanowire increased up to ~1200 nm—that is, as the distance from focal plane to the bismuth nanocrystal increased. This reflects the observation that longer nanowires are easier to trap than shorter nanowires. In the second region, the trap stiffness decreased due to a decreased magnitude of the gradient force as more of the nanowire sits outside of the focal plane.”

9) Lines 130-131: “...the gradient force experienced by the wire was independent of nanowire length.” This is only true for nanowires longer than the laser spot since the interaction region saturates (see also Simpson, Nanotechnology 2012).

The reviewer is absolutely correct. We have modified this overgeneralization in a way that is accessible to both general readers and specialists as, “As was noted during the optical trapping of similarly high refractive index nanowires in water with lengths that exceed the length of the focal volume (Rayleigh range), the gradient force experienced by the wire was independent of nanowire length.^{42,48,52} For germanium nanowires in these experiments, the gradient force acting on the germanium section of nanowire was largely responsible for enabling optical trapping, while the scattering forces experienced by the metal seed were responsible for orientation (Figure S7).⁴³”

10) Line 148: The definition of “DPG” is only given in the Supp. Info. It should be given also in the main text.

This has been corrected, and “DPG” is now defined in the main text.

Reviewer #2 (Remarks to the Author):

The major claims in this paper cover two phenomena, first is the trapping on nano-wires in non aqueous solvents and the second is the "soldering" of the nanowires together. Put together these two form what is in my opinion an important set of results that is likely to be of significant interest in the field. It is often the case that micro sized components of different materials are needed to create a device; for example various semiconductors to create light emitters of light detectors for sensing or miniature diagnostic devices. However adding these small components with optical tweezers from a water based solution is a problem due to the desire not to immerse the existing circuitry in water. The electronics community is less concerned about other solvents and oils and so the ability to optically tweeze nanowires in other media and attach them to devices will be of great interest.

As the tweezing of nanowires in different solvent is very interesting I would have expected the paper to discuss these results in relation to previously reported tweezing of colloidal particles in nonpolar solvents such as "electrostatic interactions of colloidal particles in nonpolar solvents: role of surface chemistry and charge control agents" in Langmuir 2008, 24, 1160-1164 by Dufresne et al.

We sincerely thank the reviewer for pointing out this manuscript. They are correct that it is highly relevant, and the report has already generated a wide range of interesting discussions and potential experiments. The surfaces of nanowires produced by solution-based processes are an active area of investigation, and their zeta potentials are largely underreported, limiting immediate application in this manuscript. However, we are glad to add this reference and some discussion in a few places throughout the manuscript:

On pages 4 and 5, “Although nonaqueous solvents have been explored as media for optical trapping in a few reports—for example, Spadaro *et al.* demonstrated optical trapping of gold-silica mesocapsules in ethanol,³⁴ Sainis *et al.* investigated the impact of surface moieties on organic beads in nonpolar solvents,³⁵ and Black *et al.* presented the optical trapping of silica-coated polymer microspheres in organic solvents⁸—to the best of our knowledge, the optical trapping and manipulation of colloidal inorganic nanomaterials in organic solvents has not yet been reported.”

On page 7, “When the second nanowire reached the first nanowire in the trap, it “soldered” coaxially, tip-to-tail, with the first nanowire, overcoming interfacial electrostatic and hydrodynamic forces (Figure 3d).^{28,30,35,50,}”

On page 12, “We anticipate that these results will usher in a new realm of optical trapping applications, including additive nanomanufacturing from constituent nanomaterial building blocks,⁹ single-particle catalysis,^{7,8,56} and future hydrodynamic studies.^{5,6,35,57,58,}”

The soldering of the nanowires together is also an important finding that will be of great interest to the tweezing community. The findings have good novelty however the physical mechanism that is presented could be more strongly proven as although the authors do give some good literature references and calculations that would suggest their nanowires should heat up there is little physical evidence of the alloy they claim is created.

The reviewer is correct that all of our calculations suggest that the metal-seeded nanowires heat up to an extraordinary degree in the optical trap. We agree with the reviewer that the extraordinary local heating observed in these optically trapped nanostructures will be of great interest to the community (nanosoldering being just one example). To address this, we have also included experimental measurements of these temperatures with back focal plane interferometry, further validating the temperatures predicted by theory and confirming the high temperatures of the optically-trapped nanowire (Figure 4c). Notably, these temperatures exceed the melting point of bismuth. To the significance of these temperatures, our research in this area has rapidly expanded, with several studies underway that exploit the strong local heating of these nanostructures. Among these are alternative temperature measurements of trapped nanowires using ratiometric quantum dots as luminescent reporters of local temperatures around the optically trapped nanowire. We have included the following discussion to support these new data:

On pages 8 and 9 as, “The conventional approach to determine the temperature of an optically trapped particle uses back focal plane interferometry to determine the diffusion coefficient of the particle in the trap. With sufficient knowledge of the physical characteristics of the trapping

medium and the size of the particle, this information enables calculation of the temperature of the particle.^{53,54} Using this formalism, we report both the average temperature at the hydrodynamic radius of trapped nanowires and the average temperature of the nanowire at irradiances relevant for nanosoldering (Figure 4). These average temperatures vary from 135 to 221°C at the hydrodynamic radius (defined in the SI) and 266 to 392°C at the nanowire surface, depending on the trapping power, thus illustrating significant heating of the trapped nanowire, along with a steep temperature gradient in the surrounding fluid. Notably, the temperatures at the nanowire surface are above the bulk bismuth melting point of 271°C. Moreover, to the best of our knowledge, these temperatures are higher than the temperature of any previously measured, optically trapped nanowire in solution. Finally, although this methodology used to measure particle temperature assumes that the trapped particle is isothermal, the stark differences in the optical properties of the bismuth tip and the germanium wire produce a large thermal gradient within the trapped nanowire that perturbs the local fluid properties.”

With respect to alloy formation, it is clear that the bismuth nanocrystal tip will melt once the temperature of the structure exceeds the 271°C melting temperature of pure Bi (neglecting any Gibbs-Thompson melting point depression, which would cause the tip to melt at an even lower temperature). The reviewer makes a good point that, upon melting, the composition of the liquid droplet will be almost entirely pure bismuth; however, as the temperature of the trapped nanostructure increases, the germanium content of the liquid bismuth (alloy) droplet will also increase (see the Bi-Ge phase diagram, Figure S2). Once the structure has cooled below the 271°C eutectic temperature, the Bi and Ge components will phase separate and solidify, meaning that any investigation probing the elemental composition of the liquid metal droplet on the tip of the nanowire would need to be done using an *in situ* spectroscopic probe while the structure is inside the optical trap, an investigation which is beyond the experimental reach of the current study. Notably, the Bi-Ge pair is an extraordinarily well-studied system whose phase behavior and alloy formation are critical for things like the well-established, solution-liquid-solid (SLS) seeded-growth process that was used to grow the germanium nanowires used in this study (*e.g.* Yan and Lee, 2009. *J. Phys. Chem. C.*; and Chockla *et al.*, 2011. *Chem. Mater.*). Nonetheless, as requested by the reviewer, we have removed any mention of “eutectic alloy” or “alloy composition” from the manuscript, instead simply referring to the melted Bi seeds as “liquid droplets”.

The papers conclusions would be much more convincing if they could show analysis of the soldered nanowires through Raman spectroscopic analysis, EDX, AFM or TEM to show the mix of materials created.

We include Raman spectroscopy of the dispersed germanium nanowires before laser-heating and after laser-heating in Figure 3g, however, we did not highlight that the wires retain their crystallinity after the trapping, laser heating, and assembly procedure. We thank the reviewer for pointing out that we have not sufficiently emphasized this data, for which we have now added the following text:

“Raman spectroscopy (Figure 3g) also demonstrates that ensembles of germanium nanowires dispersed in toluene retain their crystallinity even after laser heating at high irradiances with a focused ($\sim 1.5 \text{ MW}\cdot\text{cm}^{-2}$) and unfocused, collimated ($50 \text{ kW}\cdot\text{cm}^{-2}$) beam.”

and

“Raman scattering (g) from ensembles of germanium nanowires dispersed in toluene before and after photothermal heating by both a focused ($\sim 1.5 \text{ MW}\cdot\text{cm}^{-2}$) and unfocused, collimated ($50 \text{ kW}\cdot\text{cm}^{-2}$) 1070 nm laser demonstrates that photothermal heating during trapping does not alter the crystallinity of the material.”

We agree with the reviewer that EDX, AFM, and TEM would be excellent additions to the paper, providing further evidence of the final Bi-Ge nanowire heterostructure. However, to date, isolating and imaging the final heterostructures has proven to be extremely difficult and unsuccessful. Not only is isolation difficult due to the low number density of nanowires in the dispersion required for optical trapping, it is also difficult due to the high-boiling-point, viscous solvent used for optical trapping. Preparing the heterostructure for imaging with any of the suggested techniques requires a series of washes and centrifugations to remove the viscous solvent. After that, the possibility of selecting the fraction of the dispersion that contains the final heterostructure and then viewing the heterostructure on a visible section of a TEM grid becomes increasingly low. Unfortunately, despite multiple attempts, we have not yet been successful with EDX, TEM, or AFM to-date.

Reviewer #3 (Remarks to the Author):

In this manuscript, Crane et al. demonstrate optical trapping of germanium nanowires in toluene and squalene. It was observed that the nanowires orient with the long axis parallel to the Poynting vector of the trapping laser. Moreover, it was seen that trapping multiple wires results in an end-to-end “nanosoldered” product. The manuscript is well-written and the claims are largely supported by the research conducted.

I believe that TEM images of multiple soldered nanowires are necessary to demonstrate sufficient control over the nanosoldering process yielding a head-to-tail attached product.

Although we agree with the reviewer’s point that AFM and/or TEM images would be ideal to compliment optical micrographs, we have not been able to acquire either AFM or TEM images, despite multiple attempts, due to challenges with the viscous, chemical properties of the organic solvents used in these experiments. We have observed that developing new methods for direct TEM or AFM imaging of the as-made heterostructures is a substantial undertaking and, once successful, likely will require another manuscript to describe the process fully.

Acquiring TEM images of the final heterostructure has been a long-term goal of ours; however, acquiring TEM of these heterostructures has proven to be particularly difficult because these experiments are performed in a viscous, high boiling point solvent that needs to be removed before imaging with electron microscopy. At present, imaging nanowires with TEM, prior to alignment and oriented-attachment, requires multiple washes which includes a series of centrifugations and re-dispersions. Imaging the nanowire heterostructure requires a similar procedure. However, finding the single heterostructure fabricated with the laser tweezers after a sufficient washing and isolation procedure has proven to be extremely difficult. Similarly, the

likelihood of the heterostructure drying onto a visible section of the TEM grid during sample preparation for imaging is very low. Additionally, washing this particular nanowire dispersion has been difficult, as it has been intentionally diluted to improve our ability to optically trap single rods and scan the solution.

The authors reason that the germanium nanowires are always trapped with the bismuth seeded end away from the trapping laser due to the differences in complex refractive index between these two materials. This is a non-trivial claim that should be supported by an experimental observable. For example, it is not clear that the difference in scattering between bismuth and germanium is sufficient to flip the orientation of a nanowire that enters the trap with the bismuth seed towards the trapping laser.

We agree that the oriented assembly of these nanowires is a non-trivial claim and that additional evidence is important for the manuscript. In response, we have added calculations of the radiation-induced torque on the bismuth-tipped germanium nanowire as a function of nanowire length, and we have significantly modified the discussion of the optical trapping and radiation pressure-induced alignment. These results show that irradiating the nanowires produces a torque that drives alignment. This is further reflected by optical images of the soldered nanowires. In addition, analytical approximations of the trap stiffness (Figure S7) highlight that, in the optical trap, the bismuth produces a destabilizing force. We believe the combination of these data strongly support the alignment of nanowires.

On pages 6 and 7, “The disparate forces experienced by the nanowire and its metal tip are predominantly due to the large imaginary component of the refractive index of bismuth ($n_{\text{Bi}} = 3.89 - 4.33i$), as compared to germanium ($n_{\text{Ge}} = 4.65 - 0.30i$). Thus, in the optical trap, the scattering force is reduced when the bismuth sits farther from the focal plane, enabling a more stable optical trap and explaining why longer nanowires trap more efficiently (Figure S7).⁴³

These anisotropic radiation-pressure effects also occur prior to optical trapping, as the laser pushes the nanowire toward the focal plane. At large distances from the high electric field gradient in the focal plane of the optical trap, radiation pressure is the dominant force, effectively inducing a torque on the nanowire. This torque causes the nanowire to align vertically in the trap with the bismuth nanocrystal-tipped end of the wire oriented away from the laser source (Figure 2b).^{2,44,47,48} Moreover, thermophoretic^{28,49,50} effects due to anisotropic heating of the bismuth nanoparticle and hydrodynamic drag^{23,51} are additional forces that act on the nanowire before trapping to push the bismuth end of the nanowire away from the laser source, resulting in nanowire alignment.”

In Figure S7 caption as, “Figure S7. Analytical approximation of nanowire longitudinal trap stiffness as a function of length, using the theory developed by Simpson and Hanna.⁴² To include the effect of nanowire length, we used the Maxwell-Garnett approximation weighed by the volume fraction of the bismuth nanocrystal, which remains constant, and the germanium nanowire, which varies with length. In the first region, the longitudinal trap stiffness increased as the length of that nanowire increased up to ~1200 nm—that is, as the distance from focal plane to the bismuth nanocrystal increased. This reflects the observation that longer nanowires are

easier to trap than shorter nanowires. In the second region, the trap stiffness decreased due to a decreased magnitude of the gradient force as more of the nanowire sits outside of the focal plane.”

The Supplementary Movies provided by the authors demonstrate the random nature by which a nanowire approaches the optical trap. Additionally, there is also no practical way to prevent additional objects from “falling into” the optical trap (especially nanoscale objects $\sim <100\text{nm}$ that are too small to be seen in the real-time images).

The reviewer’s point that “there is no practical way to prevent additional objects from ‘falling into’ the optical trap” is a common concern for all trapping experiments, be they water-based biological trapping experiments, vacuum-based trapped-ion experiments, or experiments similar to ours. The only response to this concern is to carefully design and control the individual system, such that there are minimal adventitious objects present. In our particular case, data from the QPD shows no evidence of random nanoscale objects flying into the optical trap during the assembly process. It should also be noted that these particular demonstration experiments were performed with a higher nanowire number density than an optimized experiment would typically employ. In the long term, these challenges could be overcome through the incorporation of microfluidics or other strategies to avoid interference from unwanted objects in solution. Although our demonstration is far from being a fully optimized process, it is our hope that this result will serve as a catalyst for future research in the hierarchical assembly of nanostructures.

Finally, although the wires indeed approach the optical trap in Brownian motion with a stochastic distribution of orientations, the optical field consistently induces their ultimate coaxial alignment with the Poynting vector of the beam, thus enabling the rational oriented assembly of the constituent building blocks. As discussed in the manuscript, the viscosity of the organic trapping medium can be tuned in order to alter the kinetics of the approach, with high viscosities enabling the gradual reorientation of the nanowire as it approaches, and low viscosities leading to off-axis nanosoldering due to an insufficient timescale for reorientation. This is evidenced in the Supplementary Movies, as well as through simulations and calculations.

The practical utility of this method is also potentially hindered by the authors’ observation that optical trapped materials tend to explode when brought near the coverslip surface.

We agree that the utility of this method would be hindered by the ignition of the solvent near a coverslip interface. We should emphasize that high laser powers are not necessary for simple trapping and manipulation, so from a practical utility perspective, one can use a high laser power to facilitate the oriented nanosoldering process in solution, and then simply reduce the laser power prior to bringing the final structure into contact with a surface. In addition, one can also increase the viscosity of the solvent in order to further suppress solvent vapor bubble formation. Both of these strategies enable the safe manipulation of optically trapped materials in organic solvent systems.

Nonetheless, we believe that an example of “what not to do” is important to include from a safety perspective. The scenario of nanowire ignition was included in this manuscript primarily (i) as an important safety demonstration for what can happen when a superheated structure under

a high laser power is brought in contact with a surface in a volatile organic solvent medium (the solvent in the associated video is toluene) and (ii) as a demonstration of the remarkable degree of superheating that can be achieved by “working in solution” as opposed to “working on a substrate”, thus facilitating potential high-temperature manipulation processes in a solution-based environment.

These concerns can be assuaged if the authors can reproducibly perform the nanosoldering and isolate the products for TEM imaging.

We appreciate and agree with the author’s point that TEM would strengthen evidence of nanosoldering; however, acquiring TEM of these heterostructures has proven to be difficult because these experiments are performed in a viscous, high boiling point solvent that needs to be removed before imaging with electron microscopy. Imaging nanowires with TEM, prior to alignment and oriented attachment, requires multiple washes which includes a series of centrifugations and re-dispersions. Imaging the nanowire heterostructure requires a similar procedure. However, finding the single heterostructure fabricated with the laser tweezers after a sufficient washing procedure has proven to be extremely difficult. The probability of successfully collecting the volume of fluid in which the heterostructure is dispersed is vanishingly small. Similarly, the likelihood of the heterostructure drying onto a visible section of the TEM grid during the sample preparation for imaging is extremely low. Additionally, it is increasingly difficult to wash this particular nanowire dispersion, as it has been intentionally diluted to improve our ability to optically-trap single rods and scan the solution.

We hope that careful review of the videos provided in the Supporting Information, in addition to all of the associated measurements and calculations, will be considered convincing enough that the soldering process is occurring as hypothesized in the manuscript. The work we have submitted here opens up the possibility of many interesting future experiments in organic solvents, and we sincerely hope that either our group or another group will be able to overcome the experimental challenges associated with direct AFM or TEM imaging of optically-nanosoldered semiconductor/metal-heterostructures.

Reviewers' Comments:

Reviewer #1:

Remarks to the Author:

In my opinion the authors have addressed all the comments of the reviewers in an in-depth way. They introduced many improvements with new results. Thus, in my opinion the manuscript can be accepted for publication in Nature Communications.

Reviewer #2:

Remarks to the Author:

In their response the authors have covered all the points that were raised by the reviewers as fully as possible and I believe this paper should now be published. I understand the difficulties in obtaining TEM images of the samples and whilst it is a shame that this information is not available I don't think it is necessary to back up any claims within this revised manuscript and as such should not preclude its publication.

Reviewer #3:

Remarks to the Author:

I appreciate the changes that the authors have made in response to the other reviewers and to my comments. However, I'm afraid that the major claim made by the authors is still not backed up by any experimental evidence.

As indicated by the first three words in the title, this paper's most striking claim is oriented assembly of nanostructures. It's clear that the nanostructures fuse to create a longer wire of similar aspect ratio. But are they "soldered" in the sense of electrical conductivity? This is a weakly supported claim, but it is reasonable given the melting temperature, the calculations performed, and the power of the laser used.

In contrast, the claim of 100% tip-to-tail soldering is not at all convincing. First off, given the discussion about heating, it's not even clear if the nanostructure has reformed due to the heat of the trap and whether a concentrated bismuth tip still exists in any nanowire that has been trapped. In any case, the radial symmetry of the nanowire and the trap indicates that there are two stable orientations of the nanowire: tip up and tip down. The actual orientation adopted will result from details of the how the nanowire approaches the trap, any heat-induced nanostructure reforming, the force and torque induced on the nanowire, and the drag force of the medium. The authors provide a model that is based on the differential scattering of bismuth and germanium, but for all its complexity this model cannot predict the other data point given by the authors, namely that tip-to-tail soldering is not observed in toluene (SI: Off-Axis Nanosoldering). In fact, I would consider the authors model to be adequate if it did predict this behavior in toluene.

The authors point out that insufficient product is formed for any meaningful structural study. In my opinion, this also prevents any non-trivial claims from being made about the structure of the product. Without clear experimental characterization of the nanowire products, I do believe that there is sufficient evidence to claim that oriented soldering has been achieved.

Response to reviewers

Reviewer #1 (Remarks to the Author):

In my opinion the authors have addressed all the comments of the reviewers in an in-depth way. They introduced many improvements with new results. Thus, in my opinion the manuscript can be accepted for publication in Nature Communications.

We thank the reviewer for their thoughtful comments and critiques throughout the revision process. Their criticisms significantly improved the quality of the manuscript, and we're extremely proud of the final manuscript, which we believe will be of significant interest to the broad readership of Nature Communications.

Reviewer #2 (Remarks to the Author):

In their response the authors have covered all the points that were raised by the reviewers as fully as possible and I believe this paper should now be published. I understand the difficulties in obtaining TEM images of the samples and whilst it is a shame that this information is not available I don't think it is necessary to back up any claims within this revised manuscript and as such should not preclude its publication.

We thank the reviewer for their thoughtful comments and critiques throughout the revision process, which have significantly improved both the quality of evidence and discussion throughout the manuscript. We agree that TEM images of the samples would be very scientifically interesting and beneficial to these claims, and we look forward to further developing the methodologies to examine these assembled nanowire structures. These topics – specifically investigations of alternative, more easily separated solvents, and *in-situ* characterization – are the subject of ongoing work and will be the topics of future manuscripts.

Reviewer #3 (Remarks to the Author):

I appreciate the changes that the authors have made in response to the other reviewers and to my comments. However, I'm afraid that the major claim made by the authors is still not backed up by any experimental evidence.

As indicated by the first three words in the title, this paper's most striking claim is oriented assembly of nanostructures. It's clear that the nanostructures fuse to create a longer wire of similar aspect ratio. But are they "soldered" in the sense of electrical conductivity? This is a weakly supported claim, but it is reasonable given the melting temperature, the calculations performed, and the power of the laser used.

We thank the reviewer for their comments and recognition of the major revisions to the manuscript. The reviewer brings up an interesting question regarding the electrical conductivity of the final periodic heterostructure; however, given the difficulty we have had thus far with isolating the heterostructure for TEM, we believe we would run into similar obstacles to test the electrical conductivity of the heterostructure. We would like to emphasize that we use the word

“solder” as a description of using a low melting temperature molten metal to physically join two germanium nanorods together. The combination of images, measured temperatures, and calculated temperatures (all above the melting point of bulk bismuth) indicate that two basic events occur: first, the bismuth nanocrystal melts and forms a molten low-germanium-content alloy as it makes contact with the germanium nanowire (based on the bulk bismuth/germanium eutectic diagram); and second, the germanium nanowire-molten bismuth nanocrystal-germanium nanowire ultimately remains attached upon cooling. The combination of these measurements provides strong evidence for the soldering—that is, physical binding of different individual elements via a low melting point element—of two germanium nanowires via a bismuth nanocrystal junction.

In contrast, the claim of 100% tip-to-tail soldering is not at all convincing. First off, given the discussion about heating, it's not even clear if the nanostructure has reformed due to the heat of the trap and whether a concentrated bismuth tip still exists in any nanowire that has been trapped.

We appreciate the reviewer bringing up this concern. Solution-liquid-solid syntheses of bismuth-seeded germanium nanowires occur at temperatures in excess of 350°C [X. Lu *et al.* *J. Am. Chem. Soc.* 45, 2005] without degradation of the bismuth nanocrystal-germanium nanowire heterostructure. Thus, we do not anticipate any nanostructure degradation. Moreover, images of the nanowire heterostructures and individual nanowires before and after optical trapping show no physical change. In addition, Raman scattering from the nanowires before and after heating indicates that no appreciable chemical changes occurred after laser irradiation. As anticipated, we do not observe any evidence that the nanostructure deforms throughout the optical trapping or nanosoldering processes.

In any case, the radial symmetry of the nanowire and the trap indicates that there are two stable orientations of the nanowire: tip up and tip down. The actual orientation adopted will result from details of the how the nanowire approaches the trap, any heat-induced nanostructure reforming, the force and torque induced on the nanowire, and the drag force of the medium. The authors provide a model that is based on the differential scattering of bismuth and germanium, but for all its complexity this model cannot predict the other data point given by the authors, namely that tip-to-tail soldering is not observed in toluene (SI: Off-Axis Nanosoldering). In fact, I would consider the authors model to be adequate if it did predict this behavior in toluene.

The reviewer is correct that there are likely two stable orientations of this particular nanowire. However, the bismuth-tip-down orientation is significantly less stable due to increased scattering of the bismuth tip closer to the focal plane, which will displace the nanowire from the maximum of the gradient of the electric field, reducing the restoring gradient force. Thus, the tip-up orientation is likely more stable.

However, as the reviewer also points out, the actual orientation of the nanowire is strongly dependent on the forces the nanowire experiences as it approaches the trap, as detailed in the “Optical Trapping and Alignment in Organic Solvents” section and associated discussion. The first forces that a nanowire in Brownian motion experiences are the scattering and torque forces

that serve to both push the nanowire into the optical trap and align the nanowire into its stable orientation. As demonstrated in Figure 2, these forces always push the nanowire into the tip-up orientation. Moreover, the forces produced by the optical trap intuitively must eventually overcome the hydrodynamic forces – a fact that has been previously demonstrated experimentally [R. Agarwal *et al. Optics Express*, 13, 2015]. Thus, these forces will align the nanowire into the bismuth-tip-up orientation prior to reaching the trap. This is demonstrated by the images of nanosoldered nanowires in Figure 3, which provides conclusive evidence of optical alignment, suggesting that the model is sufficient.

The model that we present in Figure 2 includes a full finite-element solution of Maxwell's equations, including scattering and absorption, in order to predict the optical torque forces. It does not include hydrodynamic forces, which are significantly weaker than the optical trapping forces, and, thus, will not predict the off-axis nanosoldering. Thus, this lack of “prediction” is anticipated and is not evidence for a failure of the model.

The reviewer also points out that the off-axis nanosoldering from Supplementary Note 2 is insufficiently discussed. We have updated this discussion to reflect that off-axis nanosoldering occurs only occasionally in toluene. In addition, we have added another explanation for off-axis addition in low viscosity solvents. In short, low viscosity solvents produce a smaller drag force on the nanowire, and, in turn, a smaller radiation pressure is required for nanowire alignment and trapping. During a trapping event, nanowires below the focal plane within a larger solid angle are accelerated into the trap. Given the Gaussian profile of the beam, the solid angle and overall volume of nanowires that are pushed into the trap are significantly larger in toluene, due to the lower radiation pressure to overcome hydrodynamic forces, than in squalane. This can cause nanowires to enter the focal plane at larger angles in squalane resulting in off-axis nanosoldering. We thank the reviewer for inspiring this additional discussion.

The authors point out that insufficient product is formed for any meaningful structural study. In my opinion, this also prevents any non-trivial claims from being made about the structure of the product. Without clear experimental characterization of the nanowire products, I do believe that there is sufficient evidence to claim that oriented soldering has been achieved.

We absolutely agree with the reviewer that clear experimental characterization of the nanowire products would be interesting and insightful to understand the soldering process at the atomic level. As we discussed in our previous response, despite our best efforts, we were unable to isolate and characterize individual nanosoldered nanowires with AFM or TEM images due to challenges with the viscous, high boiling point organic solvents used in these experiments. Employing alternative, high viscosity, easily separated solvents and developing new methods for imaging in solution are important but substantial goals that will be the subject of future studies.

However, we think that the images of the nanosoldering process show that the nanowires are clearly welded together. As the referee wrote earlier in their review, “[nanosoldering] is reasonable given the melting temperature, the calculations performed, and the power of the laser used.” Moreover, the supplementary movies show that, after nanosoldering, the assembled nanowire can freely bend and move in solution. These observations, combined with analytical

and experimental measurements of optically trapped nanowire temperatures and simulations of optical nanowire alignment, suggest that the second nanowire aligns, heats above the melting point of bismuth, and attaches to the first nanowire in the optical trap. The quality of these heterojunctions will be the subject of future studies.